# Energy Absorption and Failure Modes of Different Composite Open-Section Crush Elements under Axial Crushing Loading

**DOI:** 10.3390/ma17133197

**Published:** 2024-06-30

**Authors:** Xulong Xi, Pu Xue, Xiaochuan Liu, Chunyu Bai, Xinyue Zhang, Xiaocheng Li, Chao Zhang, Xianfeng Yang

**Affiliations:** 1School of Aeronautics, Northwestern Polytechnical University, Xi’an 710072, China; 2020160673@mail.nwpu.edu.cn (X.X.); xuepu@nwpu.edu.cn (P.X.);; 2National Key Laboratory of Strength and Structural Integrity, Aircraft Strength Research Institute of China, Xi’an 710065, China; 3National Key Laboratory of Strength and Structural Integrity, School of Aeronautic Science and Engineering, Beihang University, Beijing 100191, China

**Keywords:** composite, failure mode, energy absorption, numerical simulation

## Abstract

In order to study the energy absorption characteristics of the open-section thin-walled composite structures with different cross-sections, axial compression tests were carried out at loading speeds of 0.01 m/s, 0.1 m/s, and 1 m/s. Finite element models were built to predict the crushing response and energy absorption behaviors of these open-section structures. The effects of the cross-section’s shape, cross-section aspect ratio, trigger mechanism, and loading speed on the energy absorption characteristics of the composite structures were analyzed. The results show that the average crushing loads of the hat-shaped and Ω-shaped open-section structures are 14.1% and 14.6% higher than those of C-shaped open-section structures, and the specific energy absorption (SEA) values are 14.3% and 14.8% higher than that of C-shaped open-section structures, respectively. For the C-shaped open-section structures, a 45° chamfer trigger is more effective in reducing the initial peak load, while a 15° steeple trigger is more appropriate for the hat-shaped open-section structures. The average crushing loads and SEA of C-shaped, hat-shaped, and Ω-shaped open-section structures are reduced when the loading speed is increased from 0.01 m/s to 1 m/s. The increase in loading speed leads to the splashing of debris and thus reduces the loading area and material utilization of open-section structures, leading to a decrease in energy absorption efficiency.

## 1. Introduction

Composite materials have become the overwhelming choice for the structural components of aircraft in recent years due to their excellent advantages in strength, weight reduction, and energy absorption behaviors [1,2,3]. For example, the number of composite materials used in Airbus A350 and Boeing 787 has exceeded 50% of their total structural weights. However, aircraft structures are inevitably subjected to impact loads in the event of a crash. In order to protect occupants from injury, the structural deformation and material failure of aircraft components may be involved to absorb the impact’s kinetic energy [4,5]. It is a critical point that aerospace composite materials possess the same crashworthiness as metal materials during an emergency landing. Thus, the energy absorption design of aerospace composite structures has always remained a top priority to prevent catastrophic structural failure and significant casualties.

Composite tubes have been increasingly used in aerospace applications. Existing studies on crashworthiness have mainly focused on evaluating and improving the crushing behavior and energy absorption capacity of various composite thin-walled structures, such as circular carbon fiber-reinforced plastic (CFRP) tubes [6,7], squared CFRP tubes [8,9], and double-hat-shaped CFRP tubes [10,11]. It was found that circular composite tubes exhibit the best specific energy absorption characteristics among all the various composite tubes with different sectional profiles, but they have the same wall thickness [12]. A large number of approaches have been proposed to enhance crashworthiness behaviors, such as filling cellular materials in composite tubes. Lightweight foams are usually used as a core material of composite tubes to improve buckling strength, stiffness, and impact resistance while maintaining minimal mass [13,14]. However, the usage of cellular materials may decrease weight efficiency, leading to undesirable specific energy absorption compared with hollow composite tubes. 

Composite materials also have disadvantages such as high cost and brittle fracture behaviors. The hybrid material design strategy provides a good solution to meet the requirements of lightweight property and energy absorption [15,16,17]. The SEA of the metal–composite hybrid structure can be improved by 37% compared to pure aluminum tubes. The external inversion mode of the metal–clad composite hybrid circular tube is the main mechanism for energy absorption [18]. In addition, it is able to avoid the excessive loads transferred to the entire structure and improve energy absorption behaviors by using the trigger mechanism [19]. Traditional composite tubes usually exhibit Euler buckling, shell buckling, and a brittle fracture mode. However, these composite tubes can collapse with a progressive crushing mode after introducing a chamfer trigger mechanism [20,21]. The crashworthiness investigation of the double- and triple-coupling triggered composite tubes reveals that the triple-coupled triggers improve the energy absorption capacity while double-coupled triggers can weaken the triggered zones to decrease the initial peak force [22]. In addition, multi-scale impact fracture mechanisms were investigated in hybrid materials [23]. 

Open-section energy-absorbing structures are extensively used as the subfloor structure of modern airplanes. Thus, a number of studies have been conducted to investigate the crushing behavior and energy absorption of open-section structures [24,25]. Bolukbasi and Laananen [26] investigated the energy-absorbing characteristics of the composite flat plates, angles, and channel sections under axial compression loading, and a semi-empirical analysis methodology was developed to predict the energy absorption capability. Riccio et al. [27] investigated the structural behavior of a channel section composite component subjected to quasi-static compression and dynamic loads. The interfaces between plies oriented according to the impact loading direction have been found more susceptible to inter-laminar damage. Jackson et al. [28] studied the crushing response of the carbon fiber/epoxy crush elements. The impact testing results indicate a 6–15% reduction in SEA compared to the quasi-static crushing case. Waimer et al. [29] investigated the dynamic failure behavior of the CFRP components assembled using a CFRP half-tube and an L-shaped strut. The assembly of the half-tube and L-strut provides an Improved section modulus, which leads to a progressive crushing mode. 

The existing studies on open-section energy-absorbing structures have focused on the crushing response with the impact velocity arranged from 1.8 m/s to 10 m/s, while it still lacks the crushing data of the composite channel section structures with different cross-sections when the impact velocity is arranged from 0.01 m/s to 1 m/s. In addition, the crushing mechanism and energy absorption performance of different composite thin-walled open-section structures have not been revealed. Accordingly, this paper aims to study the energy absorption characteristics of different open-section thin-walled composite structures using a high-speed hydraulic servo testing system. Then, a finite element model was built to further study the crushing response and energy absorption performance of open-section thin-walled composite structures. Finally, the influences of the cross-section’s shape, cross-section aspect ratio, trigger mechanism, and impact velocity on the failure modes and energy absorption characteristics of thin-walled composite structures were analyzed.

## 2. Experimental Testing

### 2.1. Specimen Description

The testing specimens are fabricated via the hot-press molding process based on CCF300/8552A high-temperature cured epoxy/carbon fiber prepregs (AVIC Composite Co., Ltd., Beijing, China). The volume fraction of the carbon fibers is approximately 67% ± 2%, and the 0° tensile strength of a single-layer prepreg is 1500 Mpa. The material density of the testing specimens is 1.6 g/cm^3^. The testing specimens are laid at angles of [45°/0°/90°/−45°/45°/0°/90°/−45°]_s_, and the subscript “s” means the symmetric lay-up method. The axial compression direction aligns with the 0° direction of the carbon fibers, and the specimen’s thickness is 2 mm. To investigate the effect of the cross-sectional configuration, cross-section aspect ratio, and triggering method on the energy absorption characteristics of composite thin-walled structures, three cross-sectional configurations are considered, including the C-type, hat-type, and Ω-type configurations, as shown in Figure 1. These geometric configurations have three different section aspect ratios, namely 1.65, 1.06, and 2.31, and the three types of cross-section aspect ratios are denoted as C1, C2, and C3, respectively. In addition, two different triggering methods, including the 45° chamfer trigger mechanisms and the 15° steeple trigger mechanisms, have been adopted to improve crushing responses. Figure 2 exhibits the testing specimens of the open-section crushing elements. All specimens have a height of 100 mm and the same cross-sectional area. In order to guarantee that these specimens are fixed in the material’s testing machine stably, a composite base along the horizontal plane is added at the bottom of the testing specimens. The composite base consists of a metal shell and a resin material with dimensions of 80 mm × 60 mm × 20 mm. The testing specimens are partially inserted into the composite base during crushing experiments.

### 2.2. Testing Setup and Procedure

The axial compression tests of composite open-section thin-walled structures were conducted using an Instron VHS 160/100-20 high-speed hydraulic servo testing machine (Instron Inc., Norwood, MA, USA), as shown in Figure 3a. This testing machine consists of a hydraulic system, water cooling system, frame, and control system. The constant dynamic loading process is realized by the loading energy provided by the combination of a hydraulic actuator and a gas accumulator. The maximum loading speed of the testing machine is 20 m/s, and the maximum impact dynamic load is 100 kN. The testing load and displacement are, respectively, measured by the force sensors and displacement sensors installed in the testing machine. The impact velocity was set at 0.01 m/s, 0.1 m/s, and 1 m/s, with a maximum compression stroke of 54 mm. The testing equipment and the specimen clamping method are illustrated in Figure 3b. The testing specimen is fixed to the base of the testing machine using two fixed blocks and bolted joints. The support can guarantee the stability of the loading device and prevent the head of the testing machine from becoming unstable during the loading process. Table 1 presents all experimental conditions of the composite open-section thin-walled structures. During the loading process, a Photron SA-X high-speed camera (Photron Inc., Tokyo, Japan), as shown in Figure 3c, was used to record the deformation and failure behaviors of the composite open-section thin-walled structures.

### 2.3. Crashworthiness Evaluation Criterion

Several crashworthiness evaluation criteria are selected to evaluate and compare the energy absorption performance of various structures [30]. Figure 4 shows a typical force-displacement curve of energy-absorbing materials during a progressive crushing process. The force–displacement curve can be classified into three typical stages: the initial crushing stage (Stage I), the stable crushing stage (Stage II), and the densification stage (Stage III). During Stage I, the crushing force shows an approximate linearity increase first and then a certain degree of decrease after reaching the peak force. During Stage II, the crushing force maintains a relatively stable interval and fluctuates around the mean crushing load. The following evaluation criteria are adopted to evaluate the energy absorption capacity of open-section crushing elements.

(1)Initial peak force *F_p_*: The maximum force during the crushing process, and *F_p_* should be controlled within an allowable range to avoid transmitting excessive loads to passengers.(2)Average crushing force *F_acf_*: The average crushing force during Stage II can be expressed as(1)$$F_{mcf}=\frac{\int Fdl}{l}$$
where *F* is the instantaneous crushing force during the crushing process, and *l* is the effective compression stroke when an energy-absorbing material is fully compacted.

(3)Crushing force efficiency *CFE*: The crushing force efficiency is defined as the ratio of the averaged load during the plateau stage to the initial peak force, which represents the load uniformity of an energy-absorbing material. It can be given by(2)$$CLE=\frac{F_{mcf}}{F_{p}}$$(4)Specific energy absorption *SEA*: It is the most significant evaluation criterion to compare the energy absorption capacity of different energy-absorbing materials, which is defined as the energy absorbed per unit mass of a structure, and it can be given by(3)$$SEA=\frac{\int Fdl}{m}$$
where *m* is the mass of the energy-absorbing material. In this work, the displacement when the head of the material testing machine contacts the open-section crushing elements is taken as the starting point for energy absorption. 

## 3. Numerical Modeling

### 3.1. Finite Element Modeling

Numerical simulations can provide more detailed insights into failure mechanisms and deformation evolution, which are not accessible via physical testing methods. This section focuses on developing a finite element model to simulate the axial crushing response of composite open-section thin-walled structures. Figure 5 shows the finite element model of composite open-section thin-walled structures under axial crushing loads. The composite laminated material is simulated using continuous shell elements (SC8R) with a mesh size of 1.5 mm. These composite specimens are composed of sixteen layers, and a two-layer unidirectional strip is set for each layer. In addition, cohesive elements are established to simulate delamination failure between the adjacent layers. A schematic diagram of the lay-up configuration and the cohesive elements is shown in Figure 6. The interlayer adhesive layer is modeled using cohesive elements (C3H8) with a thickness of 0.01 mm. The indenter is regarded as the rigid body in the finite element model. The clamping portion at the bottom of the test specimens is neglected in the numerical model. The nodes located at the bottom of the test specimens remain fixed by restricting all degrees of freedom. In order to facilitate loading and obtain the reaction forces, a reference point RP-1 is established and coupled with the indenter. The indenter moves downwards at a prescribed velocity. In addition, a general automatic contact algorithm is adopted to simulate the contact interface, the possible friction interactions between the different parts and the self-contact of the specimen are defined in the tangential contact behavior. The modeling of friction contact is based on the general contact algorithm, and its friction coefficient is set to 0.5. The hard contact relationship is adopted to define the normal contact behaviors of the model. The traction–separation model is adopted to define the cohesive contact behavior between the adjacent layers. 

### 3.2. Material Damage Modeling

(1)Intralaminar properties

In this study, the 2D Hashin failure criterion is employed to determine the failure initiation for open-section crushing elements. The Hashin criterion defines four types of failure modes, including fiber tension, fiber compression, matrix tension, and matrix compression. The expressions for these failure modes are as follows:

Fiber tension failure (*σ*_11_ ≥ 0):(4)$$F_{ft}={\left(\frac{\sigma_{11}}{X_{T}}\right)}^{2}+\alpha {\left(\frac{\sigma_{12}}{S_{12}}\right)}^{2}=1$$

Fiber compression failure (*σ*_11_ < 0):(5)$$F_{fC}={\left(\frac{\sigma_{11}}{X_{C}}\right)}^{2}=1$$

Matrix tension failure (*σ*_22_ ≥ 0):(6)$$F_{mt}={\left(\frac{\sigma_{22}}{Y_{T}}\right)}^{2}+\beta {\left(\frac{\sigma_{12}}{S_{12}}\right)}^{2}=1$$

Matrix compression failure (*σ*_22_ < 0):(7)$$F_{mc}={\left(\frac{\sigma_{22}}{Y_{C}}\right)}^{2}+\gamma {\left(\frac{\sigma_{12}}{S_{12}}\right)}^{2}=1$$

Here, *σ_ij_* (*i*, *j* = 1, 2) denotes the stress components. *X_T_* is the axial tension strength, *X_C_* is the axial compression strength, *Y_T_* is the transverse tension strength, *Y_C_* is the transverse compression strength, and *S*_12_ is the shear strength. *α*, *β*, and *γ* are scale coefficients. When *F_g_* (*g* = *f*, *m*, *t*, *c*) reaches 1, this indicates the onset of material damage in the open-section crushing elements. Here, after the initiation of failure, the material stiffness gradually degrades, and it enters the damage evolution stage, as shown in Figure 7. Point A represents the failure initiation point that satisfies the Hashin criterion, followed by linear stiffness degradation until the material failure occurs.

Once the damage initiation criterion is satisfied, a damage evolution method is required to describe its subsequent development. A linear progressive damage evolution is selected to represent the evolution process of intralaminar damage, and the damage variable *d_i_* can be given by
(8)$$d_{i}=\frac{\delta_{eq}^{t}(\delta_{eq}-\delta_{eq}^{0})}{\delta_{eq}(\delta_{eq}^{t}-\delta_{eq}^{0})}$$
where *δ_eq_* represents the equivalent displacement corresponding to the failure mode, $\delta_{eq}^{0}$ denotes the equivalent displacement at the initiation of the failure, while $\delta_{eq}^{t}$ represents the equivalent displacement when the material is fully damaged. $\delta_{eq}^{0}$ and $\delta_{eq}^{t}$ can be expressed as
(9)$$\delta_{eq}^{0}=\frac{\delta_{eq}}{\sqrt{\varphi }}$$
(10)$$\delta_{eq}^{t}=\frac{2G^{C}}{\sigma_{eq}/\sqrt{\varphi }}$$
where *φ* is the variable related to the failure mode, *G^C^* is the fracture energy density, and *σ_eq_* is the equivalent stress corresponding to the failure mode. In addition, the calculation methods of the equivalent displacement and stress for every failure mode are shown in Table 2.

In Table 2, *L_c_* represents the element’s characteristic length; the symbol “< >” is the Macaulay operator, and it can be expressed as
(11)$$<x>=\left\{\begin{array}{c} x,\;x>0\\ 0,\;x\leq 0 \end{array}\right.$$

The elasticity constants and damage initiation coefficients of the composite materials are exhibited in Table 3.

(2)Interlaminar properties

To simulate the delamination failure of the composite open-section thin-walled structures during the crushing process, the cohesive elements based on the traction–separation law are used between the layers that are prone to failure. The elastic constitutive relationships are described as follows:(12)$$\left[\begin{array}{c} \sigma_{n}\\ \sigma_{s}\\ \sigma_{t} \end{array}\right]=\left[\begin{array}{ccc} K_{n} & 0 & 0\\ 0 & K_{s} & 0\\ 0 & 0 & K_{t} \end{array}\right]\left[\begin{array}{c} \delta_{n}\\ \delta_{s}\\ \delta_{t} \end{array}\right]$$
where *σ_k_* (*k* = *n*, *s*, *t*) and *K_k_* (*k* = *n*, *s*, *t*) denote the traction stress and stiffness in the normal direction and two shear directions, and *δ_k_* is the separation displacement. The quadratic nominal stress criterion is adopted to determine the initiation of the delamination failure, and it can be given by
(13)$${\left(\frac{\langle \sigma_{n}\rangle }{\sigma_{n}^{c}}\right)}^{2}+{\left(\frac{\sigma_{s}}{\sigma_{s}^{c}}\right)}^{2}+{\left(\frac{\sigma_{t}}{\sigma_{t}^{c}}\right)}^{2}=1$$
where $\sigma_{n}^{c}$, $\sigma_{s}^{c}$, and $\sigma_{t}^{c}$ are the corresponding interface strengths of the interlaminar layer.

A power law fracture criterion is employed to predict the damage evolution of the interlaminar properties. It indicates that the delamination failure under mixed-mode conditions is governed by the power law interaction of the energies required to cause failure in the individual modes:(14)$${\left(\frac{G_{Ι}}{G_{ΙC}}\right)}^{2}+{\left(\frac{G_{ΙΙ}}{G_{ΙΙC}}\right)}^{2}+{\left(\frac{G_{ΙΙΙ}}{G_{ΙΙΙC}}\right)}^{2}=1$$
where *G*_I_, *G*_II,_ and *G*_III_ are the interface energies; and *G*_I*C*_, *G*_II*C,*_ and *G*_III*C*_ are the critical fracture energy values required to cause failure in the different modes. In addition, the material parameters of the cohesive element are given in Table 4.

## 4. Results

To validate the crushing response of the testing specimens, the C1-shaped and hat-shaped specimens were selected to establish the finite element models at a loading velocity of 1 m/s. The load–displacement curves obtained from the experimental and simulation results are shown in Figure 8. The comparisons between the peak load and the average crushing load of the testing specimens are presented in Table 5. It can be concluded that the finite element models can effectively capture the crushing response of the open-section crushing elements during the compression process. Furthermore, convergence analysis is conducted with mesh sizes of 1 mm, 1.5 mm, 2mm, and 3 mm. The simulation results shown in Figure 8c prove that 1.5 mm is an appropriate mesh size with acceptable accuracy and computational costs. In addition, the oscillatory processes are noticeable for the force–displacement curves of the specimens during the crushing process. The crushing force decreases when composite thin-walled open-section structure fractures appear. Different failure mechanisms, such as matrix cracking, fiber failure, and delamination, may be involved during the impact process. In addition, local buckling may occur, which leads to a folding sequence for composite thin-walled open-section structures; this folding phenomenon can also result in the oscillatory process.

Figure 9 exhibits the failure modes of C1-shaped and hat-shaped testing specimens with a crushing displacement of 10 mm. It is evident that both C1-shaped and hat-shaped specimens primarily experience in-plane shear deformation and interlaminar delamination during the compression failure process. For C1-shaped open-section specimens, the stress concentration at the corners leads to the formation of a relatively long axial matrix tensile failure zone. However, no significant axial matrix tensile failure zone is observed at the corners of the hat-shaped open-section specimens.

## 5. Discussions

### 5.1. The Effect of the Cross-Section Configuration

To investigate the effect of the cross-section configuration on the energy absorption characteristics of composite thin-walled structures, three geometric configurations, including the C1-shape, hat-shape, and Ω-shape, were selected. The triggering mode for all the configurations was a 45° chamfer triggering mechanism, and the loading speed was 1 m/s. Figure 10 shows the typical load–displacement curves of the open-section elements with three different cross-section configurations during the crushing process. It can be found that all three specimens exhibit a progressive crushing failure mode. The C1-shaped specimen presents a more pronounced load decrease after reaching the peak force compared with the hat-shaped and Ω-shaped specimens. In addition, the crushing force of the C1-shaped specimen at the stable crushing stage is lower than that of the other two configurations. The initial peak force, the average crushing force, and specific energy absorption of the hat-shaped and Ω-shaped specimens are almost the same with respect to the triggering mode and loading speed. The average crushing forces of the hat-shaped and Ω-shaped specimens are 14.1% and 14.6% higher than that of the C1-shaped specimen. The specific energy absorption increases by 14.3% and 14.8% for the hat-shaped and Ω-shaped specimens compared with the C1-shaped specimen, as shown in Figure 11. The relatively higher initial peak load of the hat-shaped and Ω-shaped open-section specimens can be attributed to the combined influence of the cross-section configuration and triggering mode.

During the crushing process of the testing specimens, material deformation primarily occurs through bending, delamination failure, and shear failure. The top of the testing specimen experiences bending due to interlaminar cracking, causing the inner carbon fiber fabric to bend inward and the outer carbon fiber fabric to bend outward. Simultaneously, numerous short intralaminar cracks form during the crushing process, and shear failure occurs at the root of these short intralaminar cracks, leading to the formation of numerous fragments, as shown in Figure 12.

Figure 13 exhibits the failure modes of the testing specimens with different cross-section configurations. The C1-shaped testing specimen has the largest fragment size, while the Ω-shaped testing specimen has the smallest fragment size. For the C1-shaped and hat-shaped specimens, a portion of the outer carbon fiber fabric forms a longer axial tearing region due to stress concentration at the corners. Furthermore, a portion of the inner carbon fiber fabric bends inward, and some relatively intact carbon fiber fabric remains after these specimens are fully crushed. The larger size of the remaining fragments and the presence of large intact carbon fiber fabrics in the C1-shaped specimen indicate insufficient damage during the crushing process, which is averse to energy absorption. Consequently, the C1-shaped specimen exhibits a lower average crushing force and specific energy absorption. However, no evident stress concentration zone occurs for the Ω-shaped open-section specimen during the crushing process, resulting in more complete material failure. Accordingly, the Ω-shaped specimen shows a higher average crushing force and better specific energy absorption. In addition, the stress concentration is relieved in the hat-shaped specimen due to the smoother corners; thus, these materials are fully damaged, leading to a considerable average crushing load and specific energy absorption compared to the Ω-shaped specimen.

### 5.2. The Effect of the Cross-Section Aspect Ratio

To investigate the effect of the cross-section aspect ratio on the energy absorption characteristics of thin-walled composite material structures, the crushing responses of the C1, C2, and C3 specimens were analyzed. The triggering mode of all specimens was a 45° chamfer trigger. Figure 14 shows the typical load–displacement curves for the C-shaped open-section elements with three different cross-section aspect ratios. It can be observed that the load–displacement curves for the three C-shaped open-section elements are quite similar. Figure 15 shows a comparison of the energy absorption characteristics of the testing specimens with different cross-section aspect ratios. The average crushing load of the C1 specimen is simply 6.4% higher than that of the C2 specimen and 5.1% higher than that of the C3 specimen. In addition, the SEA of the C1 specimen is 6.4% and 5.0% higher than that of the C2 and C3 specimens. The C1, C2, and C3 specimens almost have the same failure mode in that axial tearing occurs at the corners of the carbon fiber fabric. The width of the residual fragments is comparable to the specimen’s thickness, as shown in Figure 16. However, the length of the residual fragments during the crushing process is related to the specimen’s dimensions, resulting in slight differences in energy absorption characteristics for the specimens with three aspect ratios.

### 5.3. Trigger Mechanism

To investigate the influence of trigger mechanisms on the energy absorption characteristics of composite thin-walled open-section structures, the crushing responses of the C1-shaped and hat-shaped open-section specimens with the 45° chamfer trigger and the 15° steeple trigger are analyzed. Figure 17 shows the typical load–displacement curves of the C1-shaped and hat-shaped specimens with different triggering modes under axial crushing loading. Figure 18 presents a comparison of the energy absorption characteristics of the testing specimens with different trigger methods. It can be observed that the time needed to reach the peak force significantly increases for testing specimens with the 15° steeple trigger at the initial stage of the crushing process. However, no evident differences can be found for the load–displacement curves of the testing specimens with the two triggering mechanisms during the stable crushing stage. For the C1-shaped open-section specimens, the 45° chamfer trigger is more effective in reducing the peak force during the crushing process. For the testing specimens with the 15° steeple trigger, the load–displacement curve exhibits a small plateau when crushing displacement reaches approximately 4 mm. Subsequently, the crushing force continues to rise, and the slope of the load–displacement curve also increases. This behavior is attributed to the excessive weakening of the testing specimen above the corner by the 15° steeple trigger, while the weakening below the corner is insufficient, resulting in the inadequate induction of the progressive failure. 

For the hat-shaped open-section specimens, the initial peak force can be eliminated by using the 15° steeple trigger, and the load efficiency can reach 89.50%, indicating that the hat-shaped open-section specimen is well matched with the corresponding steeple trigger. Figure 19 exhibits the failure modes of the testing specimens with the steeple trigger method. It can be found that numerous material fragments occur during the crushing process of the hat-shaped specimen with the 15° steeple trigger, which indicates that more thorough material damage appears at the top of the testing specimen. In Section 5.1, the initial peak forces of the hat-shaped and Ω-shaped open-section specimens are relatively high. The main reason is that the optimal triggering mechanism may be different compared to composite thin-walled open-section structures with different cross-section configurations. The 45° chamfer trigger does not sufficiently weaken the hat-shaped and Ω-shaped specimens, which is not the optimal triggering mode for this loading condition.

### 5.4. Impact Velocity

To study the effect of the loading rate on the energy absorption performance, the crushing responses of C1-shaped, hat-shaped, and Ω-shaped open-section specimens at the impact velocities of 0.01 m/s, 0.1 m/s, and 1 m/s are investigated in this section. Figure 20 shows the typical load–displacement curves for the three types of testing specimens with different impact velocities. Figure 21 presents a comparison of the energy absorption characteristics of testing specimens with different impact velocities. The initial peak force, average crushing force, and specific energy absorption of three types of the testing specimen decrease as the impact velocity increases. When the loading speed is increased from 0.01 m/s to 1 m/s, the average crushing force of C1-shaped, hat-shaped, and Ω-shaped open-section specimens decreases by 6.1%, 10.9%, and 6.1%, respectively. However, the specific energy absorption of these testing specimens decreases by 6.2%, 11.0%, and 6.2%, respectively. Figure 22 shows the variation in the specific energy absorption of these testing specimens with different impact velocities. It can be concluded that the hat-shaped specimen experiences a more pronounced decline in specific energy absorption with an increase in impact velocity compared with the C1-shaped and Ω-shaped open-section specimens.

Figure 23 shows the failure modes of the testing specimens with a loading speed of 0.01 m/s, which were observed by comparing the failure behaviors of the testing specimens at loading speeds of 0.01 m/s and 1 m/s. A few material fragments from the testing specimens with a loading rate of 0.01 m/s appear during the crushing process, indicating that a thorough material failure occurs for this loading condition. The fracture morphology of the testing specimens reveals that the end section of the crushed zone retains more material fragments and exhibits more chaotic and twisted morphology, as shown in Figure 24. The crushed zone experiences adequate friction and compaction with the indenter, which can contribute to an increased structural load-bearing area. However, the testing specimens with the loading rates of 1 m/s ejected a large number of material fragments during the crushing process, reducing the load-bearing area and material utilization. Furthermore, frictional energy absorption was observed between the indenter and the crushed zone, as well as between the layers and fragments. Thus, the initial peak load, the average crushing load, and the specific energy absorption of the specimens also decrease with an increase in impact velocity.

## 6. Conclusions

This work mainly studies the crushing response and energy absorption characteristics of composite open-section thin-walled structures with different geometric configurations and loading conditions. The main conclusions obtained are as follows:(1)For the composite thin-walled open-section structures with different geometric configurations, a brittle failure mode can be observed under different loading rates. The impact kinetic energy is mainly absorbed via material bending, delamination failure, shear failure, and friction between the crushing zone during the crushing.(2)The cross-section configuration significantly influences the energy absorption characteristics of composite thin-walled open-section structures. The insufficient material damage caused by stress concentration is the main reason for the low energy absorption efficiency of C1-shaped specimens compared to the hat-shaped and Ω-shaped specimens.(3)Different triggering mechanisms primarily affect the initial crushing stage of the composite structures, while it has little influence on the stable crushing stage. For C-shaped specimens, a 45° chamfer trigger yields better energy absorption. However, the 15° steeple trigger is the optimal triggering mode for the hat-shaped structures.(4)The average crushing force and specific energy absorption of composite thin-walled structures decrease with an increasing in loading rates. More material fragments can be ejected under higher loading rates, which reduces the structural load-bearing area, material utilization, and frictional energy absorption in the crushing zone.

## Figures and Tables

**Figure 1 materials-17-03197-f001:**
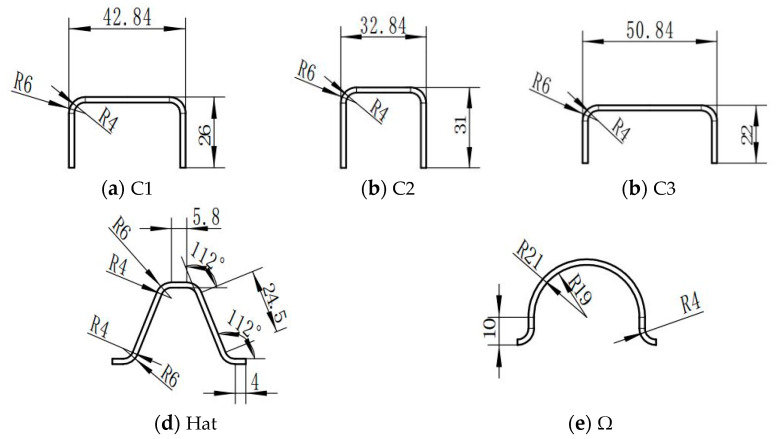
The cross-section dimensions of the testing specimens.

**Figure 2 materials-17-03197-f002:**
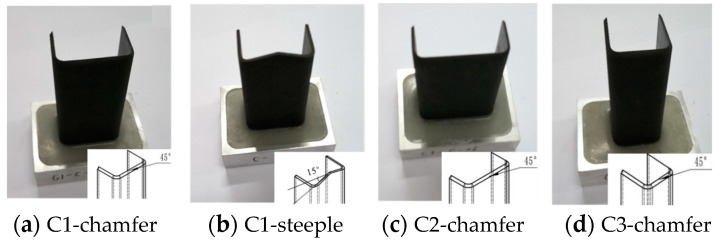
The testing specimens of the open-section crushing elements.

**Figure 3 materials-17-03197-f003:**
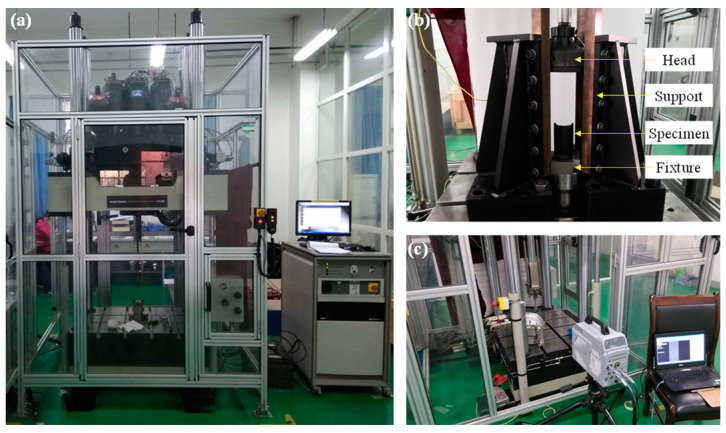
Experimental setup for the dynamic crushing tests: (**a**) INSTRON VHS 160 high-speed servo-hydraulic testing machine; (**b**) installation and schematic of the specimens; (**c**) Photron SA-X high-speed camera.

**Figure 4 materials-17-03197-f004:**
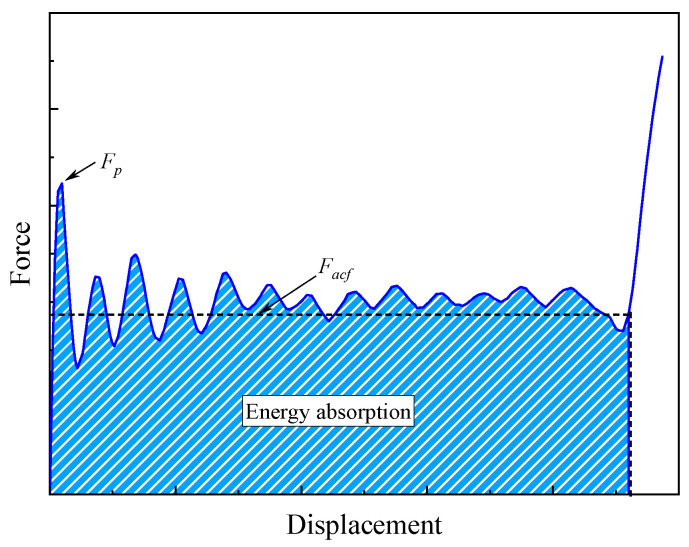
A typical load–displacement curve of an energy-absorbing material.

**Figure 5 materials-17-03197-f005:**
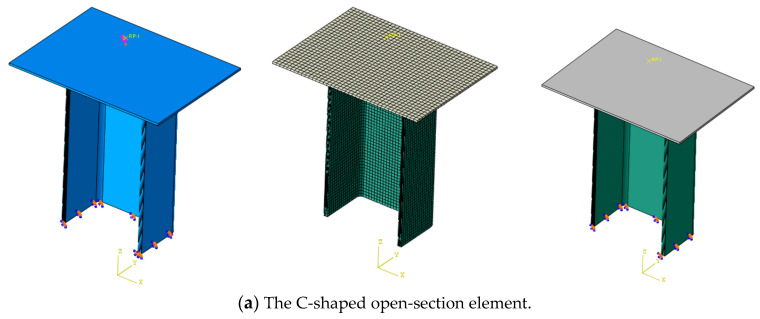
Finite element models of the open-section crushing elements.

**Figure 6 materials-17-03197-f006:**
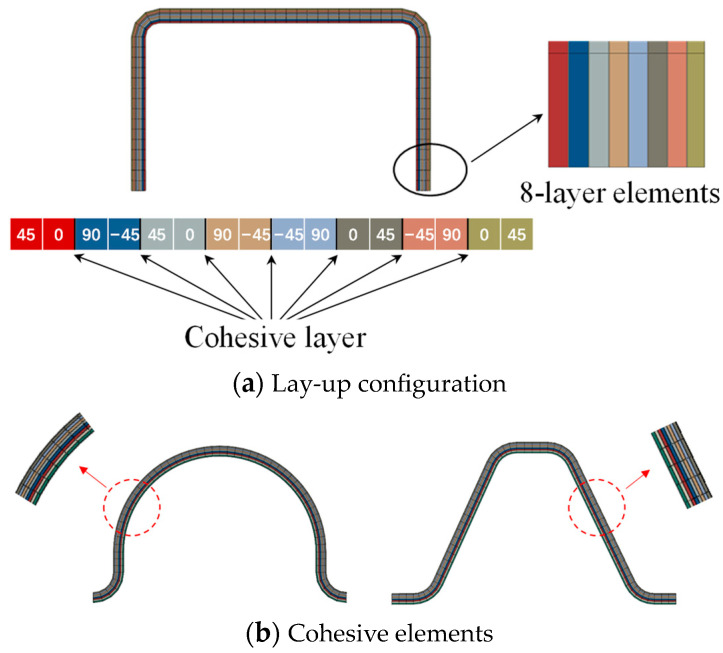
Detailed modeling method for the open-section crushing elements.

**Figure 7 materials-17-03197-f007:**
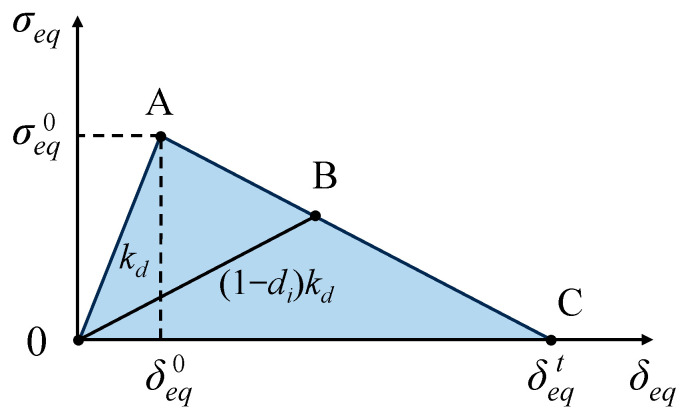
Constitutive model and damage evolution model.

**Figure 8 materials-17-03197-f008:**
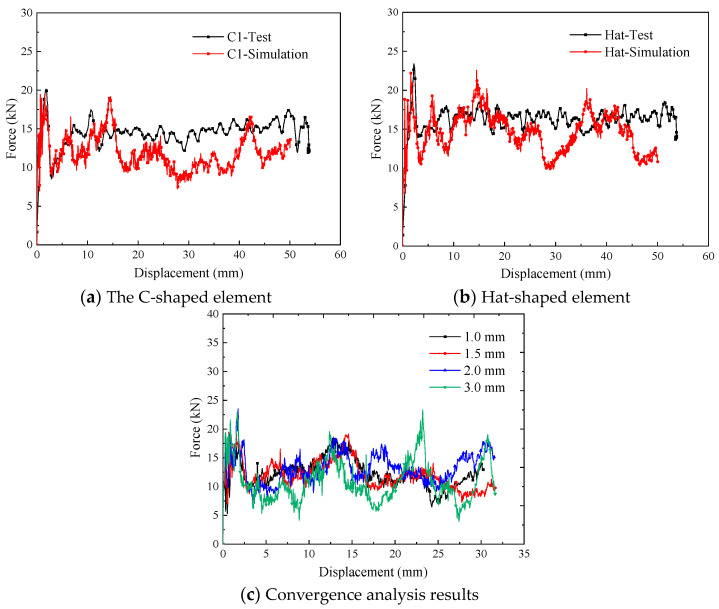
Comparison of simulation and experimental results for the crushing process.

**Figure 9 materials-17-03197-f009:**
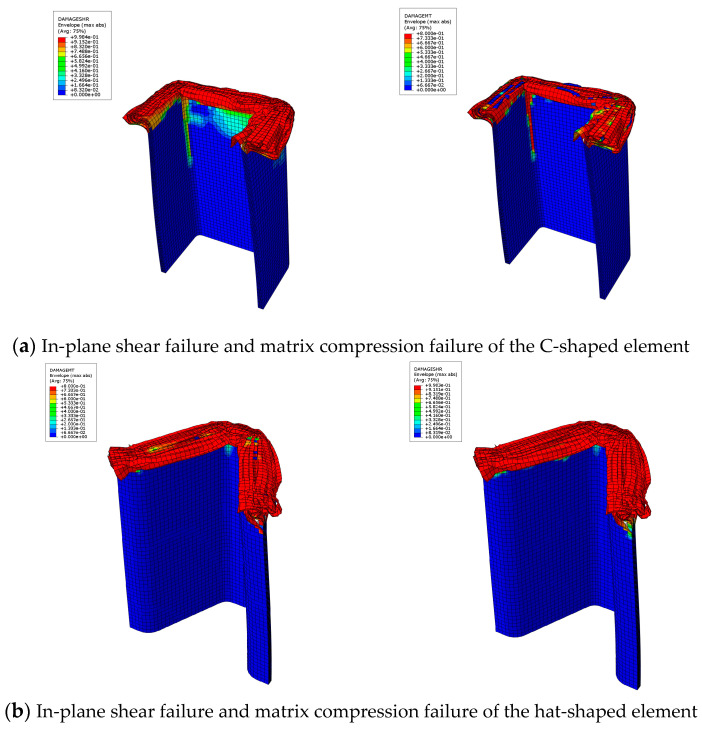
Simulation results of the failure modes of different open-section crushing elements.

**Figure 10 materials-17-03197-f010:**
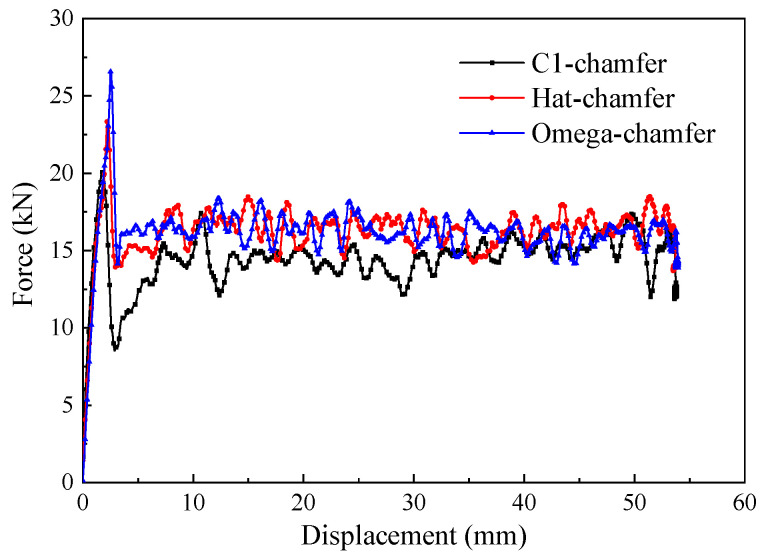
Typical force–displacement curves of the testing specimens with different cross-section shapes.

**Figure 11 materials-17-03197-f011:**
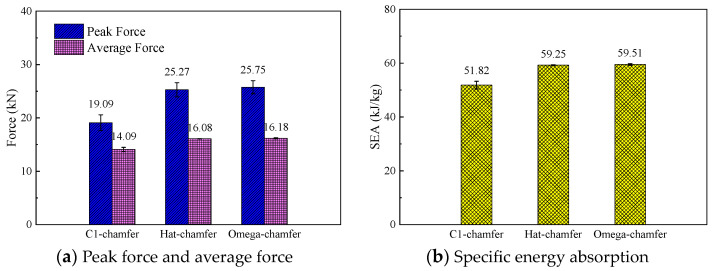
Comparison of the energy absorption characteristics of the testing specimens with different cross-section shapes.

**Figure 12 materials-17-03197-f012:**
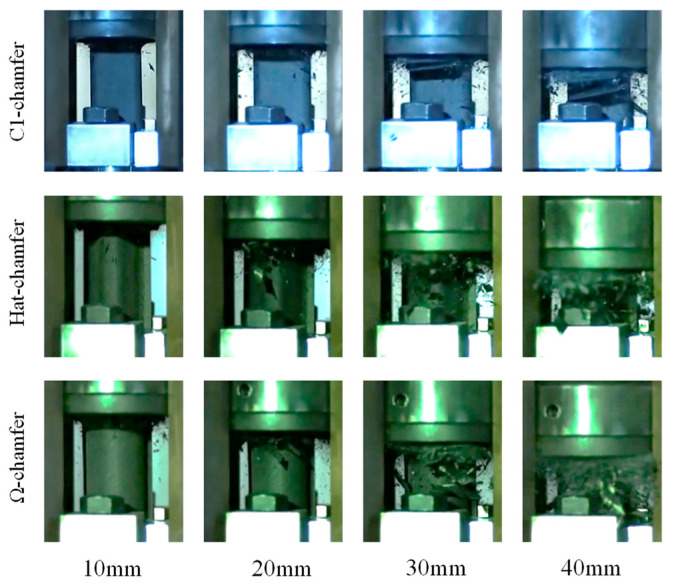
Crushing process of the testing specimens with different cross-section configurations.

**Figure 13 materials-17-03197-f013:**
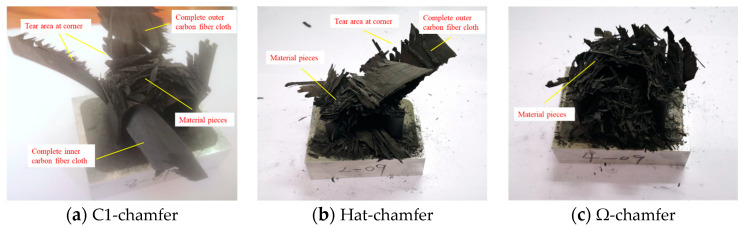
Failure modes of the testing specimens with different cross-section configurations.

**Figure 14 materials-17-03197-f014:**
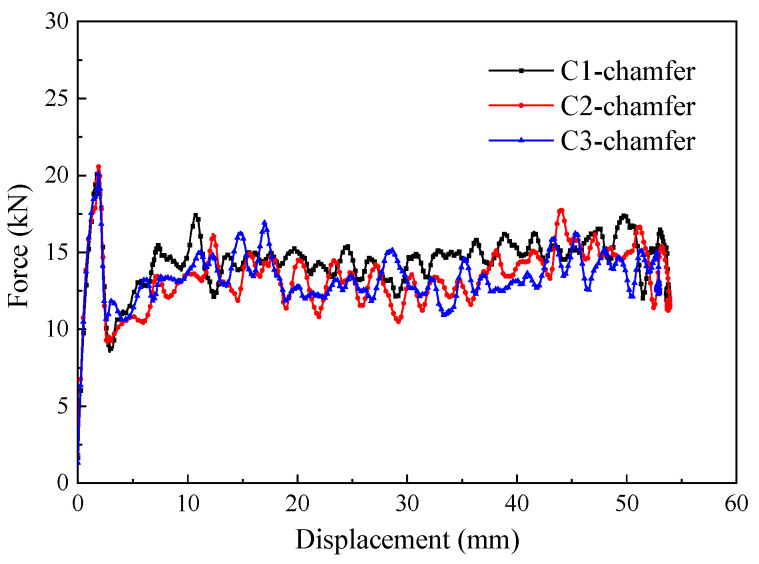
Typical force–displacement curves of the testing specimens with different cross-section aspect ratios.

**Figure 15 materials-17-03197-f015:**
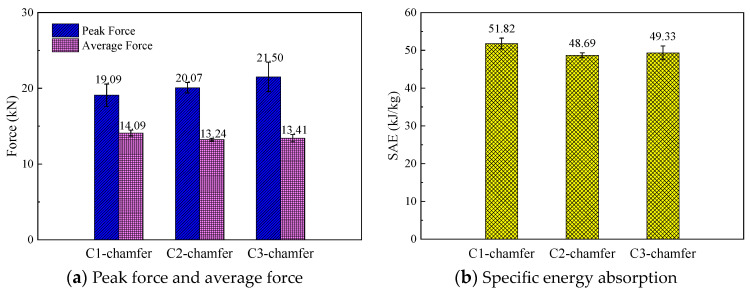
Comparison of the energy absorption characteristics of the testing specimens with different cross-section aspect ratios.

**Figure 16 materials-17-03197-f016:**
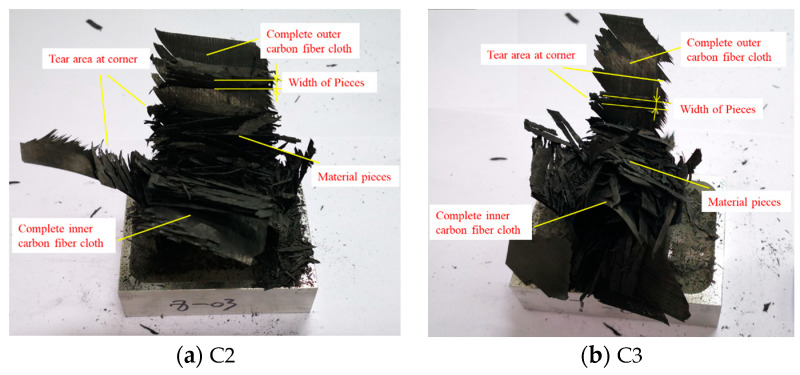
Failure modes of the testing specimens with different cross-section aspect ratios.

**Figure 17 materials-17-03197-f017:**
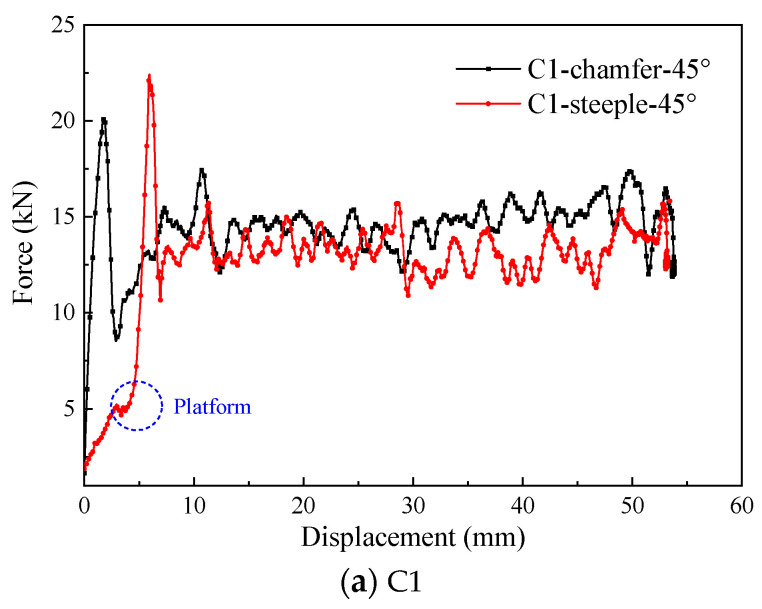
Typical force–displacement curves of the testing specimens with different trigger mechanisms.

**Figure 18 materials-17-03197-f018:**
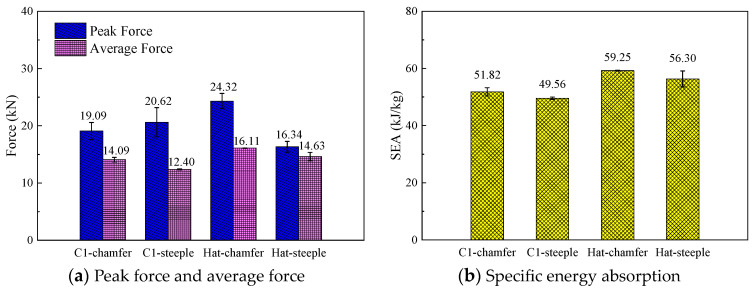
Comparison of the energy absorption characteristics of the testing specimens with different trigger mechanisms.

**Figure 19 materials-17-03197-f019:**
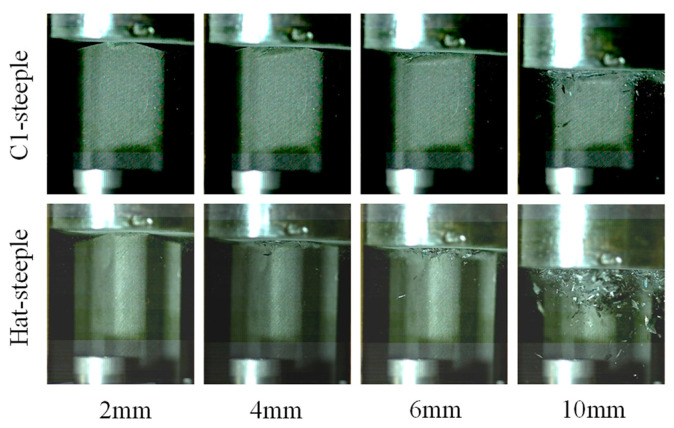
Failure modes of the testing specimens with steeple triggers.

**Figure 20 materials-17-03197-f020:**
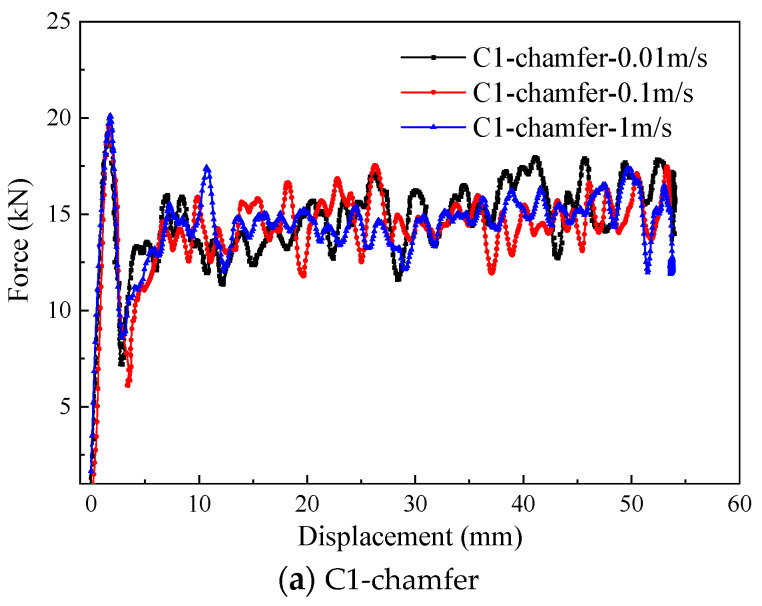
Typical force–displacement curves of the testing specimens with different loading rates.

**Figure 21 materials-17-03197-f021:**
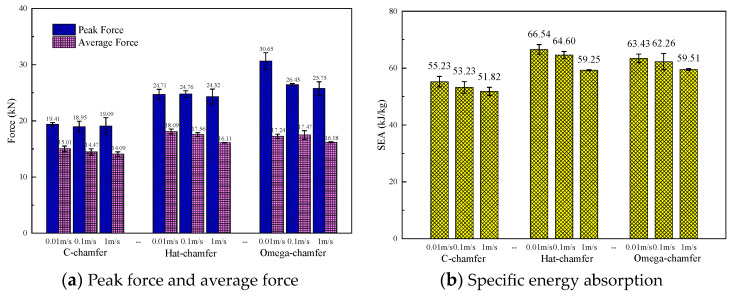
Comparison of the energy absorption characteristics of testing specimens with different loading rates.

**Figure 22 materials-17-03197-f022:**
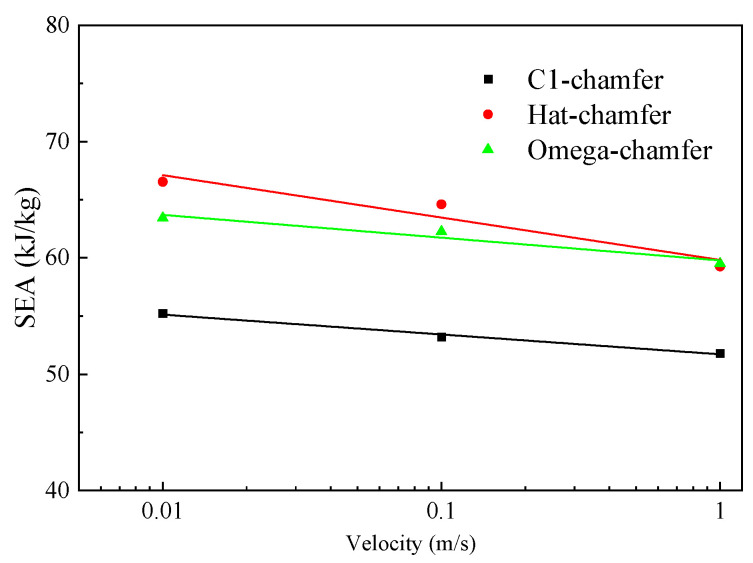
Variation in the SEA of testing specimens with different loading rates.

**Figure 23 materials-17-03197-f023:**
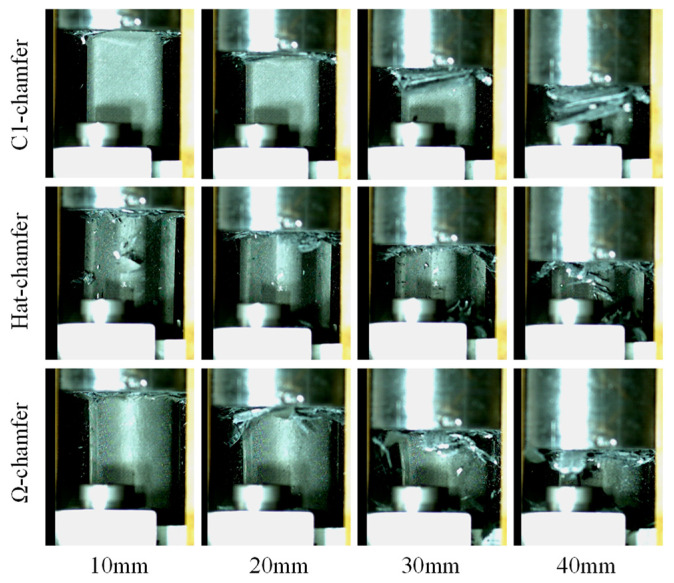
Failure modes of the testing specimens with a loading speed of 0.01 m/s.

**Figure 24 materials-17-03197-f024:**
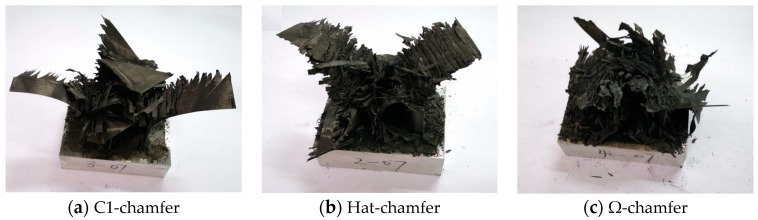
Failure modes of the testing specimens with a loading speed of 0.01 m/s.

**Table 1 materials-17-03197-t001:** Experimental conditions of composite open-section thin-walled structures.

Configuration	Triggering Mechanism	Impact Velocity (m/s)	Mass (g)
C1	45° Chamfer trigger	1.00	14.680
Hat	45° Chamfer trigger	1.00	14.678
Ω	45° Chamfer trigger	1.00	14.677
C2	45° Chamfer trigger	1.00	14.680
C3	45° Chamfer trigger	1.00	14.680
C1	15° steeple trigger	1.00	13.507
Hat	15° steeple trigger	1.00	14.029
C1	45° Chamfer trigger	0.01	14.680
C1	45° Chamfer trigger	0.10	14.680
Hat	45° Chamfer trigger	0.01	14.678
Hat	45° Chamfer trigger	0.10	14.678
Ω	45° Chamfer trigger	0.01	14.677
Ω	45° Chamfer trigger	0.10	14.677

**Table 2 materials-17-03197-t002:** Equivalent displacement and stress for every failure mode.

Failure Mode	Equivalent Stress	Equivalent Displacement
Fiber tension	$\sigma_{ft,eq}=\frac{L_{c}\left(\langle \sigma_{11}\rangle \langle \varepsilon_{11}\rangle +\sigma_{12}\varepsilon_{12}\right)}{\delta_{ft,eq}}$	$\delta_{ft,eq}=L_{c}\sqrt{{\langle \varepsilon_{11}\rangle }^{2}+{\left(\varepsilon_{12}\right)}^{2}}$
Fiber compression	$\sigma_{fc,eq}=\frac{L_{c}\langle -\sigma_{11}\rangle \langle -\varepsilon_{11}\rangle }{\delta_{fc,eq}}$	$\delta_{fc,eq}=L_{c}\langle -\varepsilon_{11}\rangle$
Matrix tension	$\sigma_{mt,eq}=\frac{L_{c}\left(\langle \sigma_{22}\rangle \langle \varepsilon_{22}\rangle +\sigma_{12}\varepsilon_{12}\right)}{\delta_{mt,eq}}$	$\delta_{mt,eq}=L_{c}\sqrt{{\langle \varepsilon_{22}\rangle }^{2}+{\left(\varepsilon_{12}\right)}^{2}}$
Matrix compression	$\sigma_{mc,eq}=\frac{L_{c}\left(\langle -\sigma_{22}\rangle \langle -\varepsilon_{22}\rangle +\sigma_{12}\varepsilon_{12}\right)}{\delta_{mc,eq}}$	$\delta_{mc,eq}=L_{c}\sqrt{{\langle -\varepsilon_{22}\rangle }^{2}+{\left(\varepsilon_{12}\right)}^{2}}$

**Table 3 materials-17-03197-t003:** Elasticity constants and damage initiation coefficients for the intra-laminar model [31].

Description	Variable	Value
Longitudinal Young modulus	*E* _1_	171,420 MPa
Transversal Young modulus	*E* _2_	9080 MPa
Principal Poisson’s ratio	*v* _12_	0.32
Shear modulus	*G* _12_	5290 MPa
Longitudinal tensile strength	*X_T_*	1773 MPa
Longitudinal compressive strength	*X_C_*	1264 MPa
Transversal tensile strength	*Y_T_*	62.3 MPa
Transversal compressive strength	*Y_C_*	199.8 MPa
In-plane shear strength	*S* _12_	92.3 MPa
Longitudinal tensile fracture energy	$G_{ft}^{\;c}$	120 N/mm
Longitudinal compressive fracture energy	$G_{fc}^{\;c}$	100 N/mm
Transverse tension fracture energy	$G_{mt}^{\;c}$	2 N/mm
Transverse compression fracture energy	$G_{mc}^{\;c}$	5 N/mm

**Table 4 materials-17-03197-t004:** Material parameters of the cohesive elements [32].

Description	Variable	Value
The stiffness in the normal direction	*K_n_*	1 × 10^6^ N/mm^3^
The stiffness in the first shear direction	*K_s_*	5 × 10^5^ N/mm^3^
The stiffness in the second shear direction	*K_t_*	5 × 10^5^ N/mm^3^
The interface strength in the normal direction	$\sigma_{n}^{c}$	60 MPa
The interface strength in the first shear direction	$\sigma_{s}^{c}$	110 MPa
The interface strength in the second shear direction	$\sigma_{t}^{c}$	110 MPa
Mode-I fracture toughness	*G* _I*C*_	0.2 N/mm
Mode-II fracture toughness	*G*_II*C*_, *G*_III*C*_	5 N/mm

**Table 5 materials-17-03197-t005:** Initial peak forces and mean crushing forces of different testing specimens.

Description	*F_P_* (Kn)	*F_MCF_* (Kn)
C1-Test	20.14	14.36
C1-Simulation	19.53	11.90
Error	3.01%	17.10%
Hat-Test	23.38	16.13
Hat-Simulation	22.57	14.65
Error	3.46%	9.18%

## Data Availability

The original contributions presented in the study are included in the article, further inquiries can be directed to the corresponding author.

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
