# Peer review of "Energy Absorption and Failure Modes of Different Composite Open-Section Crush Elements under Axial Crushing Loading"

_materials, 2024, doi:10.3390/ma17133197_

Round 1

Reviewer 1 Report

Comments and Suggestions for Authors

The development of new experimental methods for studying the processes of deformation and fracture of composite thin-walled open-section structures under impact loadings based on the combined use of modern testing equipment and calculation methods is an important scientific task. In the article, the authors studied the crush response and energy absorption characteristics of open-section thin-walled composite structures with various geometric configurations and loading conditions. The main patterns of energy absorption by these open-section structures have been established based on axial crushing tests, as well as numerical modelling. The influence of the cross-sectional shape, cross-sectional aspect ratio, trigger mechanism, and loading speed on the energy-absorbing characteristics of composite structures was assessed. The results obtained are valuable for science and practice, but there are several questions:

1. I propose to further substantiate formula (3). All existing criteria assume energy dissipation over the area of the sample being destroyed. And in (3) mass appears. Is this correct?

2. How was the accuracy of the modelling performed assessed?

3. During the fracture of composite thin-walled open-section structures, oscillatory processes are noticeable. Perhaps it would be worthwhile to determine their stages structure, as well as to more deeply interpret and describe their physical nature?

4. In the article, in fact, the process of fracture is discussed only at the macro level. At the same time, it is much more complicated. It might be worth using the multiscale approach proposed by Professor Stukhlyak: https://link.springer.com/article/10.1134/S1029959915010075

I propose to consider Stukhlyak’s method in the Introduction and justify the approach chosen by the authors of this article.

Reviewer 2 Report

Comments and Suggestions for Authors

Presented manuscript is well organized and carefully written. Problem is practically important. I propose publishing the manuscript but after some improvements. New, original elements should be underlined in introduction and conclusion points. The wider range of literature should be analysed. For example the following works should be examined: Behaviour of composite columns of closed cross-section under in-plane compressive pulse loading,  R. Mania, K. Kowal-Michalska, Thin-Walled Structures 45 (2007) 902–905; Experimental tests of stability and load carrying capacity of compressed thin-walled multi-cell columns of triangular cross-section, M. Krolak, K. Kowal-Michalska, R. Mania, J. Swiniarski, Thin-Walled Structures 45 (2007) 883–887.

The FEM model should be presented in more detail. The program, which was used should be written. The convergence analysis or other approach to check the model should be presented.

The reference to presented material parameters should be written. 

Reviewer 3 Report

Comments and Suggestions for Authors

materials-3052252:

Follow my comments:

*English must be improved.

*The abstract must be shorter; only present the main conclusions.

*The keywords are too long. Use words not phrases.

*1. Introduction: The introduction must be short and must be scientific. As it stands, the section is a textbook (very didactic). Authors must present the frontier of knowledge and the contributions of the paper to advance knowledge. Fully review the introduction. Show the importance of the paper for the area.

*2.1 Specimen description: Why did the authors choose this type of sample? justify.

Furthermore, it must be explained why the geometries in Figure 1 were chosen. Would there be more part section geometries? What is the standard used to construct the test specimens of the profiles?

*2.2 Testing setup and procedure: The standard for the test was not mentioned. The use of different loading speeds was also not justified.

*2.3 Crashworthiness evaluation criterion: There are several criteria available in the literature; Why did the authors choose these? They must justify. Who developed these criteria? The authors do not present citations. Review the document and include author citations.

*3.1 Finite element modeling: include author citations.

*5. Discussions: The authors discuss the results very well. However, they do not compare with existing literature; with other criteria; with other models for damage; etc. This aspect is missing in the discussion of the results.

*6. Conclusions: very long; reduce!

Comments on the Quality of English Language

Moderate editing of English language required.

Round 2

Reviewer 1 Report

Comments and Suggestions for Authors

Accept.

Reviewer 3 Report

Comments and Suggestions for Authors

materials-3052252R1:

A good review was carried out.

Comments on the Quality of English Language

Minor editing of English language required.